# A systems-level framework for drug discovery identifies Csf1R as an anti-epileptic drug target

Prashant K. Srivastava[1], Jonathan van Eyll [2], Patrice Godard [3], Manuela Mazzuferi[2], Andree Delahaye-Duriez [1,4,5], Juliette Van Steenwinckel[5], Pierre Gressens[5,6], Benedicte Danis[2], Catherine Vandenplas[2], Patrik Foerch[2], Karine Leclercq[2], Georges Mairet-Coello[2], Alvaro Cardenas[2], Frederic Vanclef[2], Liisi Laaniste[1], Isabelle Niespodziany[2], James Keaney[2], Julien Gasser[2], Gaelle Gillet[2], Kirill Shkura[1], Seon-Ah Chong[2], Jacques Behmoaras[7], Irena Kadiu[2], Enrico Petretto [8,9], Rafal M. Kaminski [2] & Michael R. Johnson[1]

The identification of drug targets is highly challenging, particularly for diseases of the brain. To address this problem, we developed and experimentally validated a general computational framework for drug target discovery that combines gene regulatory information with causal reasoning ("Causal Reasoning Analytical Framework for Target discovery"—CRAFT). Using a systems genetics approach and starting from gene expression data from the target tissue, CRAFT provides a predictive framework for identifying cell membrane receptors with a direction-specified influence over disease-related gene expression profiles. As proof of concept, we applied CRAFT to epilepsy and predicted the tyrosine kinase receptor Csf1R as a potential therapeutic target. The predicted effect of Csf1R blockade in attenuating epilepsy seizures was validated in three pre-clinical models of epilepsy. These results highlight CRAFT as a systems-level framework for target discovery and suggest Csf1R blockade as a novel therapeutic strategy in epilepsy. CRAFT is applicable to disease settings other than epilepsy.

[1] Division of Brain Sciences, Imperial College London, London W12 0NN, UK. [2] UCB Pharma, Avenue de l'industrie, Braine-l'Alleud R9, B-1420, Belgium. [3] Clarivate Analytics (formerly the IP & Science Business of Thomson Reuters), 5901 Priestly Drive, #200, Carlsbad, CA 92008, USA. [4] UFR de Santé, Médecine et Biologie Humaine, Sorbonne Paris Cité, Université Paris 13, Bobigny, France. [5] PROTECT, INSERM, Sorbonne Paris Cité, Université Paris Diderot, Paris, France. [6] School of Biomedical Engineering & Imaging Sciences, Centre for the Developing Brain, King's College London, St. Thomas' Hospital, London SE1 7EH, UK. [7] Centre for Complement and Inflammation Research, Imperial College London, London W12 0NN, UK. [8] Duke-NUS Medical School, Centre for Computational Biology, 8 College Road, Singapore 169857, Republic of Singapore. [9] Faculty of Medicine, MRC Clinical Sciences Centre, Imperial College London, London W12 0NN, UK. These authors contributed equally: Prashant K. Srivastava, Jonathan van Eyll. Correspondence and requests for materials should be addressed to E.P. (email: enrico.petretto@duke-nus.edu.sg) or to R.M.K. (email: rafal.kaminski@ucb.com) or to M.R.J. (email: m.johnson@imperial.ac.uk)

Despite advances in our understanding of disease processes at the molecular and cellular levels, modern drug discovery has failed to deliver improved rates of approval for mechanistically novel drugs[1]. One reason for the high rate of attrition in drug development, particularly for diseases of the central nervous system, is inadequate target validation in early-stage drug discovery[1,2]. Optimism that advances in gene discovery would facilitate the validation of mechanistically novel drug targets has yet to materialize, and there is a requirement for new approaches to target discovery and validation.

Network-based systems analyses provide powerful techniques for elucidating molecular processes and pathways underlying disease[3–7]. The power of the gene network approach comes from the analysis of multiple genes in functionally enriched pathways, as opposed to traditional single gene approaches that examine only one component of a complex system at a time. Using genome-wide transcriptional profiling in tissues relevant to the disease under investigation, gene co-expression network analysis can identify modules (i.e., sets of co-expressed genes) as candidate regulators and drivers of disease states. Network-based drug discovery aims to harness this knowledge to identify drugs capable of restoring the expression of disease modules toward health[8,9]. At this systems level, therapeutic compounds are judged not by their binding affinity to a particular protein, but by their ability to induce a transcriptional response (i.e., a gene expression profile) that is anti-correlated to the coordinated transcriptional program underpinning the disease state. This systems approach to disease modification is loosely termed the "signature reversion paradigm" and is orthogonal to traditional concepts of drug discovery.

Currently, drugs capable of reversing disease-related transcriptional signatures are identified using two main strategies. One approach uses public databases of transcriptomic profiles of cell lines treated with chemical compounds and seeks a chance anti-correlated overlap between a drug's gene expression profile and a disease's gene expression signature[10]. Whilst successful examples for the use of such "perturbation databases" have emerged[11,12], new targets are not identified by this route and the method's reliance on chance overlap in expression profiles means a very large number of drugs may need to be profiled to find one with a suitable profile[13]. A second approach has therefore emerged that aims to map the underlying drivers and regulators of disease-related gene expression signatures as candidate drug targets[7]. Successful examples include mapping the upstream regulators of disease modules using expression quantitative trait loci mapping[14,15], and approaches that make use of regulatory interactions between genes ("regulomes")[16]. As currently formulated, however, these approaches also have limitations. For example, expression quantitative trait loci mapping of networks may identify only large genomic regions in which several candidate genes could be equally implicated whilst regulome approaches are not currently formulated to specifically identify regulators that have tractability as drug targets.

Given these constraints we aimed to develop a new framework for drug target discovery based on identifying the regulators of disease-associated gene co-expression modules. Our method, "Causal Reasoning Analytical Framework for Target discovery" or CRAFT, combines gene regulatory information with a causal reasoning framework to computationally predict cell surface receptors with a direction-specified influence over module activity. We specifically chose to develop a method connecting module expression to membrane receptors because more than half of all approved drugs target receptors[17], thus maximizing the opportunity for drug repositioning and rapid experimental medicine proofs of principle. Although in this study we applied CRAFT to epilepsy, CRAFT is equally applicable to any disease for which an underlying disease expression signature can be identified.

We chose to study epilepsy using CRAFT for two main reasons. Firstly, epilepsy is a highly debilitating disease of the brain for which there is a global unmet need—approximately one in three epilepsy patients are resistant to all currently available antiepileptic drugs (AEDs) and none of the current drugs are disease modifying or curative[18]. Secondly, epilepsy is a disease benefiting from well-characterized pre-clinical models with proven relevance to the human disease, and indeed, pre-clinical testing in rodent models of epilepsy remains the mainstay for determining efficacy of candidate antiepilepsy drugs[19]. This attribute allowed the development and validation of CRAFT in a controlled experimental framework.

Epilepsy itself is a disease characterized by recurrent unprovoked epileptic seizures, but is also associated with additional features including cognitive and behavioral impairments and a heightened risk of death[20]. The causes of epilepsy can be broadly divided into cases that arise through no cause other than a genetic predisposition ("genetic epilepsy"), and epilepsy which develops secondary to an acquired brain injury such as following status epilepticus (SE) or head injury ("acquired epilepsy")[21]. Here, focusing on acquired epilepsy, we set ourselves the challenge of developing, implementing, and validating a computational method for inferring drug targets for epilepsy from disease-related gene expression data.

## Results

**Identification of candidate gene networks for epilepsy.** A summary and description of the study workflow is shown in Fig. 1. As a first step, we aimed to identify gene co-expression networks (i.e., modules) associated with epilepsy. To this end, we used an established post SE mouse model of acquired temporal lobe epilepsy (TLE)[22]. In this model of epilepsy the mice develop spontaneous recurrent seizures approximately 4 weeks after pilocarpine-induced SE. As well as manifesting spontaneous epileptic seizures, these mice also reflect several of the behavioral and cognitive disturbances associated with TLE in humans, and their response to AED therapy has been shown to be predictive of drug efficacy in human epilepsy[23].

High-throughput sequencing of mRNA (RNA-sequencing (RNA-seq)) was performed on whole hippocampus samples from 100 outbred epileptic mice and 100 control (i.e., pilocarpine-naïve) littermate mice (see Methods). In total, 14,188 genes were expressed (Log$_2$ FPKM >0) in at least 5% of samples, and of these, 9013 genes showed significant (false discovery rate (FDR) <0.05) differential expression (DE) between epileptic and healthy control mice (Supplementary Data 1).

To identify gene co-expression modules related to epilepsy, the set of genes expressed in the mouse epileptic hippocampus samples were first clustered according to their co-expression relationships. Briefly, Spearman's rank correlation coefficients of expression were computed for all gene pairs and the pairwise correlation coefficients were used to perform hierarchical clustering based on Ward's method[24]. The optimal number of co-expression modules was calculated using Elbow's and pseudo F-index method[25] (Supplementary Figure 1). This led to the identification of 28 co-expression modules and an additional gene set (termed "module" 3) consisting of un-clustered genes (see Supplementary Data 2 for the full list of modules and their constituent genes). The 28 co-expression modules varied in size between 78 and 1036 genes (mean and median module size was 255 and 188 genes, respectively).

Analysis of the biological terms and canonical pathways enriched among the 28 co-expression modules in the mouse

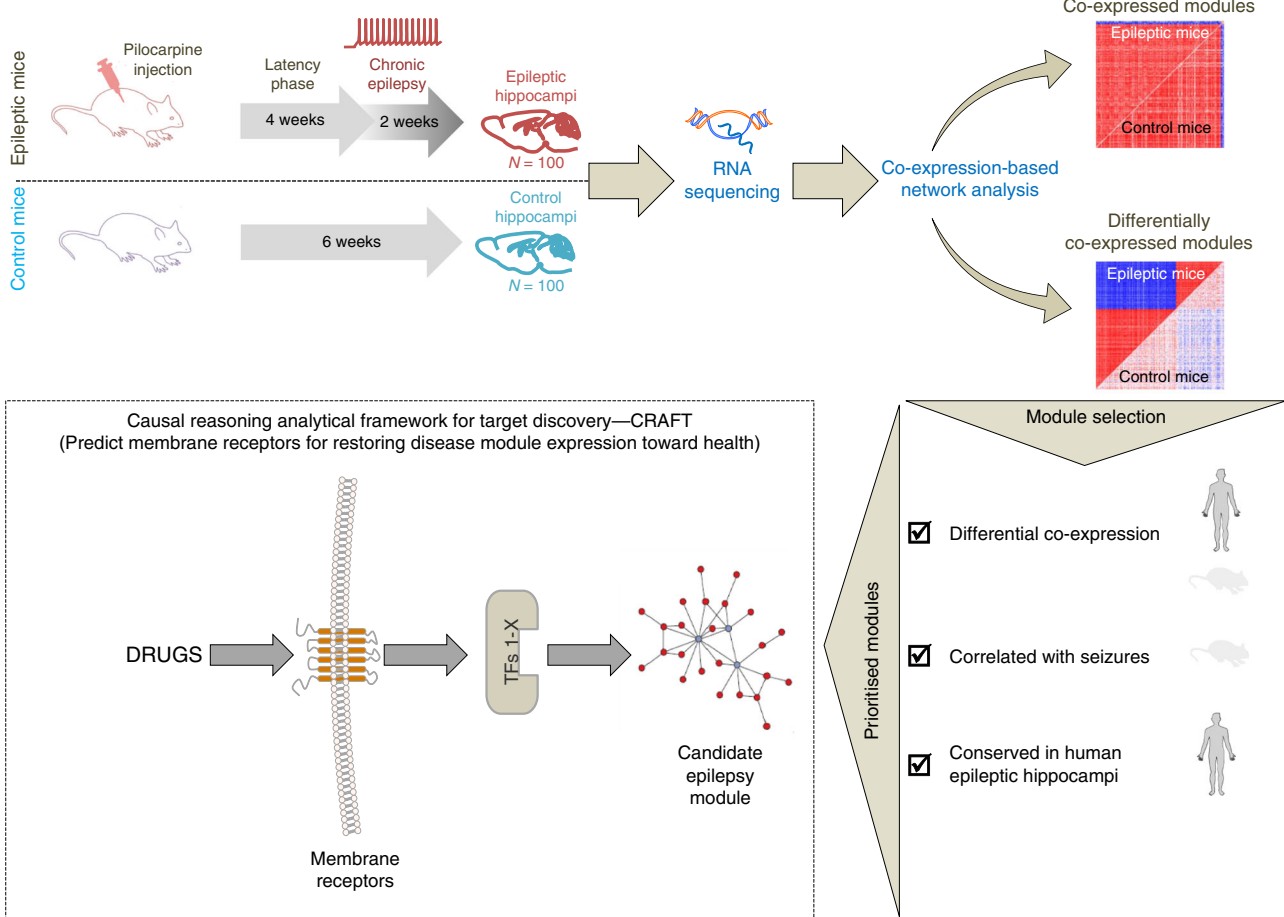

**Fig. 1** Experimental plan and study overview. We studied 100 mice with epilepsy (pilocarpine post status epilepticus model of temporal lobe epilepsy) and 100 control (pilocarpine-naïve) matched littermate mice. At 4 weeks post status epilepticus, each mouse was continuously monitored using 3D accelerometry and video monitoring for 14 days to record seizure frequency and severity. High-throughput mRNA sequencing (RNA-seq) was generated using RNA from snap-frozen whole hippocampus samples from the mice and gene expression profiles were used to generate co-expression modules. Co-expression modules with a potential relationship to epilepsy were prioritized using the following criteria: (i) differential co-expression between epileptic and healthy hippocampus (mouse and human TLE), (ii) correlation of module expression with seizure frequency (mouse), and (iii) conservation in the human epileptic hippocampus. Modules meeting these criteria were considered candidate modules for epilepsy, and subjected to CRAFT analysis to identify membrane receptors predicted to restore disease module expression toward health

epileptic hippocampus revealed that the modules were generally enriched for specific functions—the top Gene Ontology (GO) biological processes enriched in each module are shown in Supplementary Figure 2a. The results of the functional enrichment analysis for each module are reported in full in Supplementary Data 3. Among the modules with overlapping functions, modules 5, 16, and 18 were enriched for "immune response" processes (Benjamini–Hochberg (BH)-corrected $P = 2.1 \times 10^{-11}$, $P = 1.4 \times 10^{-6}$, and $P = 1.3 \times 10^{-33}$, respectively), and modules 10, 14, 26, and 29 were enriched for neuronal functions including "synaptic transmission" (BH $P = 4.4 \times 10^{-11}$, $P = 0.02$, $P = 4.0 \times 10^{-3}$, and $P = 4.0 \times 10^{-4}$, respectively).

To provide insights into the cell-type expression of the modules, we used cell-type marker genes derived from single-cell RNA-seq analysis of the mouse hippocampus (see Methods)[26]. The individual modules demonstrated notable cell-type specificity (Supplementary Figure 2b and Supplementary Data 4). The cell-type specificity of a module broadly corresponded to its functional enrichment. For example, "immune response" modules 16 and 18 were enriched for microglia marker genes, whilst "synaptic transmission" modules 10, 14, 26, and 29 were specific for neuronal cell types.

To prioritize modules with a potential relationship to epilepsy, we undertook a staged set of analyses summarized in Supplementary Figure 3. First, we tested if any of the modules were specific to the epileptic hippocampus using differential co-expression analysis. The differential co-expression paradigm postulates that a disease is linked to co-expression patterns that are different in disease compared to healthy states, reflecting perturbed functional processes. Using methodology formulated by Choi and Kendziorski[27] (see Methods), 12 modules were found to be significantly (FDR <0.05) differentially co-expressed between the epileptic and control hippocampus, whilst 16 modules displayed conservation of co-expression (Supplementary Figure 4 and Supplementary Data 5). Of the 12 differentially co-expressed modules, modules 5, 16, and 18 were enriched for functional terms related to immune response processes, whilst modules 8, 10, and 21 were enriched for synaptic transmission and/or neuronal plasticity. As expected, the 16 modules with similar co-expression patterns in epileptic cases and controls were generally enriched for "housekeeping" functional terms unrelated to epilepsy, such as cell morphogenesis and protein transport.

To further prioritize the modules in terms of their relationship to epilepsy, we explored the correlation between each module's

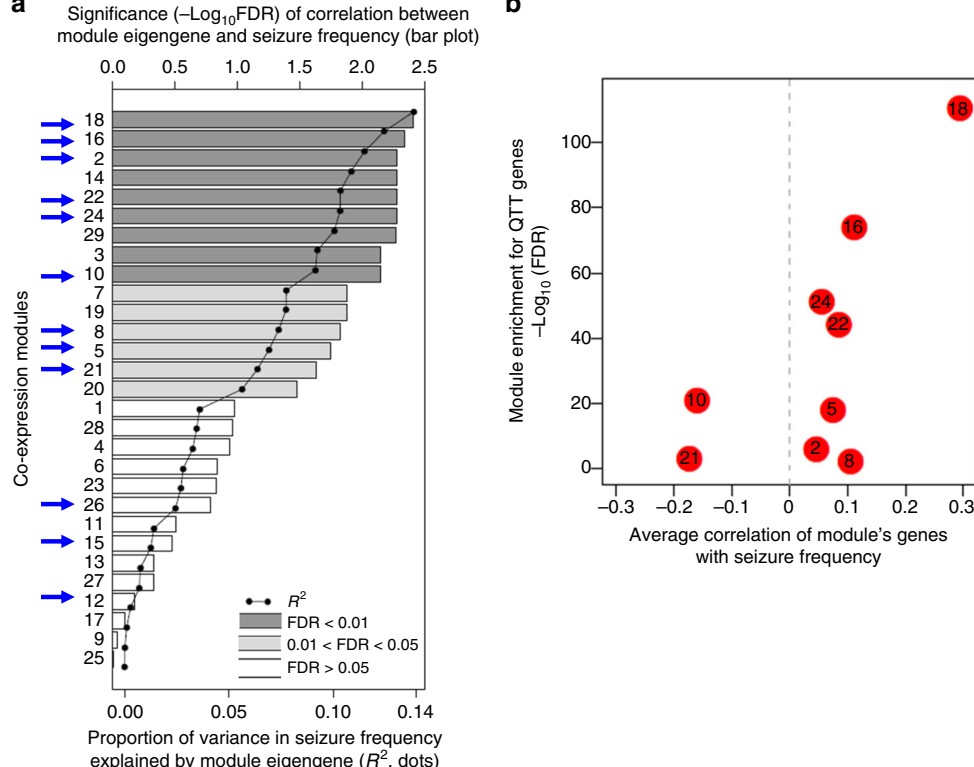

**Fig. 2** Correlation of module expression with seizures. **a** For each co-expression module from the epileptic mouse hippocampus, we plotted the significance ($-\text{Log}_{10}$ FDR) of the Spearman's correlation between the module's eigengene and seizure frequency (bar plot), and the percentage of variance in seizure frequency explained by the module's eigengene ($R^2$, dotted line). Modules marked with a blue arrow are the modules differentially co-expressed between the epileptic mouse hippocampus and the control mouse hippocampus. Modules highlighted in gray (bar plot) are significantly (FDR <0.05) correlated with seizure frequency. **b** Volcano plot of average (Spearman's) correlation of a module's genes with seizure frequency (X-axis) versus the significance of the module's enrichment for genes individually correlated with seizure frequency (QTT genes) (Y-axis) for the nine modules differentially co-expressed in epilepsy and correlated with seizures by module eigengene

expression and seizure frequency. To this end, we first quantified the frequency of behavioral seizures in each epileptic mouse by 14 days of continuous motion sensing 3D accelerometry synchronized with continuous video monitoring (see Methods). This revealed a diurnal variation of seizure occurrence in mice as well as clustering of seizures reflective of the classical patterns of human TLE (Supplementary Figure 5)[28]. Next, we summarized each module's expression by its eigengene (i.e., its first principal component, PC1) and calculated the correlation between each module's eigengene and seizure frequency. Of the 12 differentially co-expressed modules, nine (modules 2, 5, 8, 10, 16, 18, 21, 22, and 24) had an eigengene that significantly (FDR <0.05) correlated with seizure frequency (Fig. 2a).

To explore the relationship between a module and epilepsy in more detail, we correlated the expression of each individual gene in a module with seizure frequency using quantitative traits transcript (QTT) analysis[29]. Across all modules, 833 genes had expression levels that significantly (FDR <0.05) correlated with seizure frequency (hereon termed "QTT genes") (Supplementary Data 7). To assess the overall direction of correlation between a module and epileptic seizures, we plotted the average correlation of expression of genes in a module with seizure frequency against the module's enrichment for QTT genes (Fig. 2b). Considering the nine modules differentially co-expressed in epilepsy and correlated with seizures by module eigengene (i.e., modules 2, 5, 8, 10, 16, 18, 21, 22, and 24), module 18 (enriched for inflammatory processes and expressed in microglia) was the

module most significantly positively correlated with seizures, whilst module 10 (synaptic transmission) was the module most significantly negatively correlated with seizures. This anti-correlation between down-regulated modules enriched in synaptic functions and up-regulated modules enriched in inflammatory microglial pathways has also been described in autism spectrum disorder[30].

For the nine modules differentially co-expressed in epilepsy and correlated with seizures (i.e., modules 2, 5, 8, 10, 16, 18, 21, 22, and 24), we then assessed whether the module was conserved in the human epileptic hippocampus. Using human orthologs of mouse module genes and genome-wide gene expression data from 122 human epileptic hippocampus samples surgically ascertained from TLE patients[14], we found that all nine modules were conserved in the human epileptic hippocampus (FDR <0.05) (Supplementary Data 6). The conservation of these nine modules across human and mouse TLE provides an independent line of evidence for the validity of these modules, and further supports the relevance of the pilocarpine post SE mouse model of TLE to human TLE[14].

As a final assessment of the relationship of these nine mouse TLE modules to human epilepsy, we tested whether each module was also differentially co-expressed in human TLE. In this analysis, for each module, we compared intra-module correlations in the human epileptic hippocampus with that in the non-diseased human hippocampus using post-mortem hippocampal samples ascertained from people with no history of psychiatric or

neurological disease (see Methods)[31]. Among the nine mouse modules differentially co-expressed in epilepsy and correlated with seizures, seven (5, 10, 16, 18, 21, 22, and 24) were also differentially co-expressed in human TLE (Supplementary Data 6). These seven modules were selected for further analysis. Specifically, we hypothesized that focusing on these seven modules (and by extension their enriched functional pathways) would provide a starting point for the development of new therapies for epilepsy.

Before proceeding to mapping the upstream regulators of these modules as candidate drug targets for epilepsy, since an important goal of our study was to identify mechanistically new drugs for epilepsy, we asked whether any of the seven modules could be considered to have a "known" relationship to epilepsy based on the published biomedical literature (see Methods). Briefly, we first extracted published Abstracts for every gene in the genome using SCAIview webserver (www.scaiview.com) (3,811,179 abstracts with at least one gene–Abstract pair). The weight of evidence relating a particular gene to epilepsy was then quantified by determining if that gene's co-citation with epilepsy (23,092 Abstracts with at least one gene–epilepsy co-citation) was more frequent than expected by chance (hypergeometric test, Supplementary Data 8). Then, by considering gene–epilepsy pairs significant at FDR <0.05, the modules were ranked according to their enrichment of gene–epilepsy pairs (hypergeometric test, Supplementary Data 9). Of the seven candidate epilepsy modules prioritized above, only module 10 (enriched for neuronal processes) was significantly (FDR <0.05) enriched for genes with a "known" relationship to epilepsy, suggesting the remaining modules may be capturing novel functional relationships with epilepsy.

**Drug target prioritization through causal reasoning (CRAFT).** The above analyses prioritized seven modules (5, 10, 16, 18, 21, 22, and 24) as candidate modules for epilepsy by virtue of being (a) differentially co-expressed in mouse and human TLE, (b) conserved across mouse and human TLE, and (c) correlated with seizure frequency. From the pragmatic perspective of drug discovery, we set out to identify regulators of each of these modules as potential antiepilepsy drug targets.

According to the signature reversion paradigm, if a module's expression is causally related to the disease, then restoration of the disease module's expression toward the healthy state should be predictive of therapeutic benefit. We therefore set out to identify a drug-able target capable of restoring the activity of one or more candidate epilepsy module toward health. Since approximately 60% of existing drugs in clinical use target membrane receptors[17], we decided to focus our search on finding membrane receptors exerting a regulatory influence over module activity. To this aim, we developed and implemented a computational approach that combines "causal reasoning" with gene regulatory information to rank receptors based on the strength of their predicted effect on module expression (see Methods). Briefly, using Clarivate Analytics MetaBase® (version 6.15.62452), we first extracted information relating to known interactions between membrane receptors and transcription factors (TFs) via linear canonical pathways and then between TFs and their target genes. To provide context to this "interactome," only membrane receptors, TFs, and target genes expressed in the mouse hippocampus were considered (resulting in a list of 1624 expressed TFs and 307 expressed receptors). In a causal reasoning framework (logic summarized in Fig. 3), there are multiple scenarios by which a membrane receptor can act via TFs on the set of genes in a module that are dysregulated in disease. For each of these scenarios, the direction of effect of a

membrane receptor on TFs and of the TFs on target genes is defined by a causal reasoning argument, which takes into account the directionality of the receptor > TF > target gene interactions and whether the network genes are over-expressed or under-expressed in the disease state. For each scenario (Fig. 3), the significance of the influence of a membrane receptor on a module's gene expression can be quantified by considering the overlap between the direction-specified receptor effects on gene expression with the genes in a module that are over-expressed or under-expressed in epilepsy (hypergeometric test; see Methods and Supplementary Figure 6). This process allows membrane receptors to be ranked in terms of their predicted effect on module expression and the direction of that effect in terms of either activating or repressing the disease state, allowing the therapeutic directionality of receptor blockade or activation to be inferred.

Of the seven candidate epilepsy modules, four (5, 16, 18, and 22) were significantly (FDR <0.05) enriched for one or more direction-specified receptor effect on module expression (Supplementary Data 10) (for intermediate TF effects on module expression, see Supplementary Data 11). For each of these receptors, we plotted the proportion of genes in a module targeted by the receptor against the module's $-Log_{10}$ FDR enrichment for receptor (direction-specified) target genes, allowing membrane receptors to be visualized in terms of their predicted directionality on the genes in a module which are over-expressed or under-expressed in disease, as well as the specificity and magnitude of the predicted effect (Supplementary Figures 7a–d). In support of the validity of our causal reasoning results, we found that membrane receptors related to interleukin-1 type 1 receptor and Toll-like receptor 4 had a predicted direction of effect on epilepsy via a module enriched for relevant functional processes that was in agreement with the previously reported experimental evidence for that receptor[32].

Of the many membrane receptors predicted to significantly influence the expression of module genes in a direction-specified manner, macrophage colony-stimulating factor (M-CSF) receptor (also known as colony-stimulating factor 1 receptor encoded by the *Csf1R* gene in the mouse) was predicted to be a regulator of two of the seven prioritized candidate epilepsy modules (modules 18 and 22, $P = 0.017$ and $P = 0.031$, respectively). For both these modules, Csf1R was predicted by CRAFT to "activate" the subset of genes in the module that are over-expressed in epilepsy. According to the CRAFT causal reasoning framework (Fig. 3), small molecule blockade of Csf1R should therefore be therapeutic in epilepsy (i.e., reduce seizures) and restoration of module expression toward health by Csf1R inhibition should be predictive of therapeutic benefit. The availability of the known Csf1R inhibitor PLX3397[33] provided us with a tool compound by which to experimentally test this hypothesis. Moreover, Csf1R has not previously been linked to epilepsy, allowing the opportunity for novel target discovery. We therefore chose to prioritize Csf1R for further analysis.

**Csf1R regulates module 18 genes.** To test the predicted regulatory influence of Csf1R on modules 18 and 22, we first selected three genes in each module as markers of module expression (*Emr1, Aif1, Irf8, Gfap, ItgA5,* and *Serpine1*) on the basis that these genes (a) belonged to either module 18 or 22 and were among the set of genes predicted by CRAFT to be positively regulated by Csf1R, (b) are over-expressed in epileptic cases compared to controls, and (c) are not predicted to be regulated by c-Kit (a kinase also inhibited by PLX3397)[34]. Epileptic mice were treated with PLX3397 at 3 or 30 mg/kg per day or vehicle for 7 days (see Methods). Hippocampus RNA

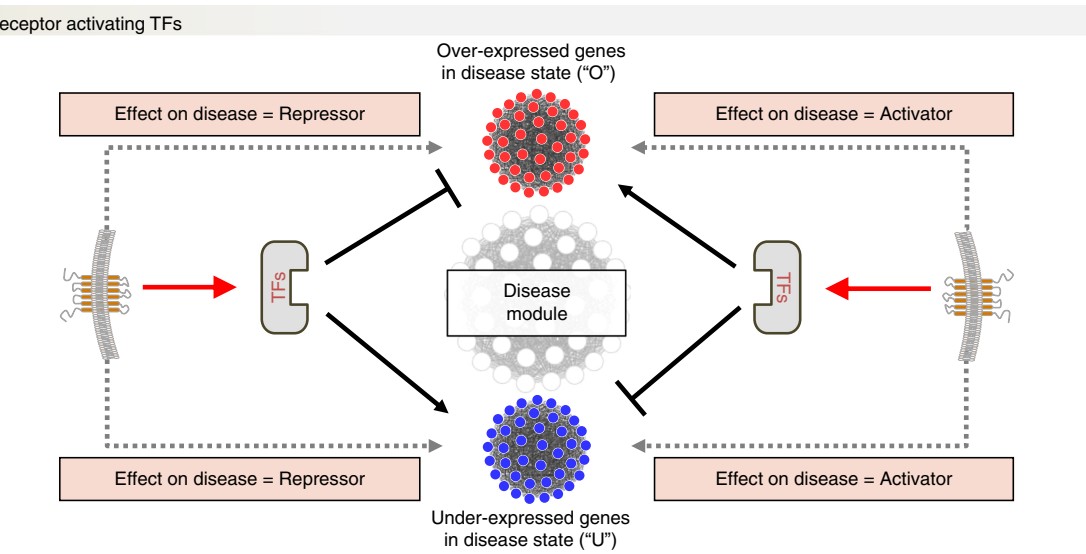

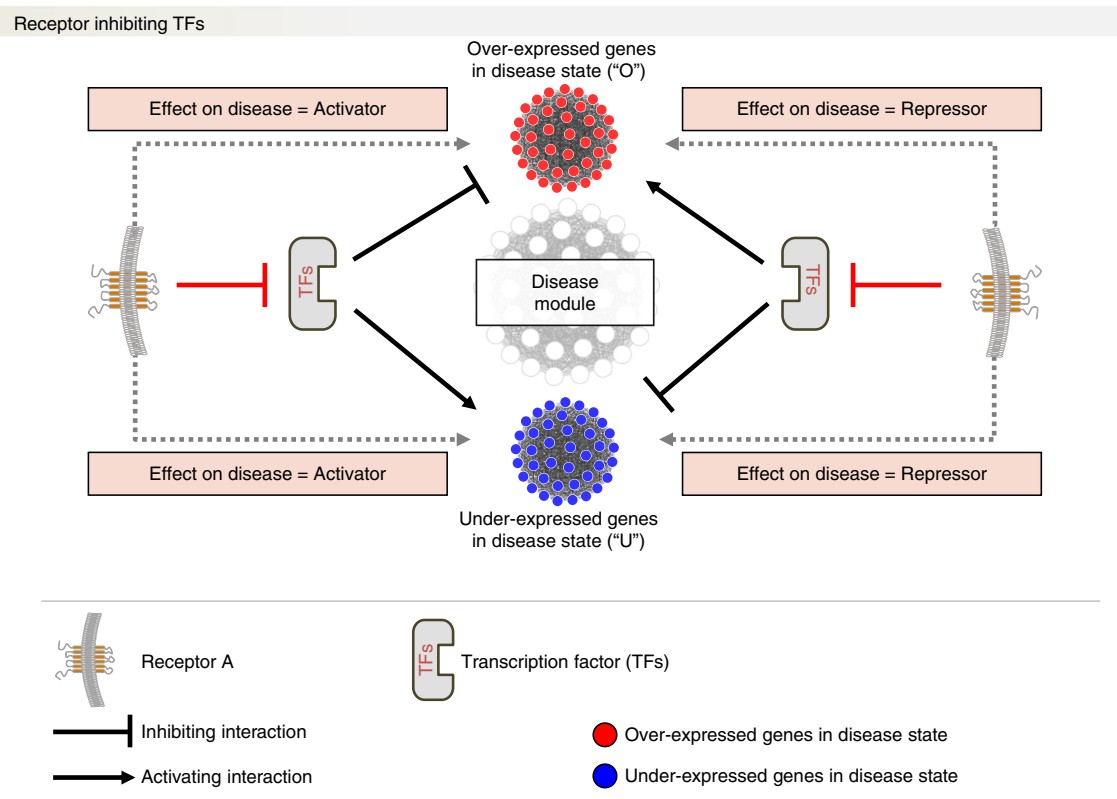

**Fig. 3** Causal reasoning framework. A knowledge-based "interactome" (or "regulome") connecting membrane receptors to module gene expression was constructed based on experimentally validated connections between membrane receptors and transcription factors (TFs) in linear pathways, and between TFs and their target genes (genome-wide). This "regulome" is then integrated with information about whether the genes in a candidate module are over-expressed ("O") or under-expressed ("U") in the disease state, allowing receptors to be classified as either disease "Activators" or "Repressors," which in turn permits the therapeutic directionality of receptor blockade or activation to be inferred. In the upper part of the figure, we show the positive activation of the TFs by the receptor "Receptor A," whereas in the lower part of the figure we show the opposite scenario of inactivation of the TFs by Receptor A. An illustrative example of the framework is shown in Supplementary Figure 6

was extracted on day 7 and gene expression was measured by reverse transcriptase quantitative PCR (qPCR). Csf1R blockade with PLX3397 at 30 mg/kg per day was associated with a significant decrease in the marker genes of module 18 but not module 22 (Fig. 4a).

To confirm the regulatory influence of Csf1R on module 18, and to investigate the transcriptional response of module 18 to

Csf1R blockade in more detail, we assayed the expression of all 171 genes in module 18 in response to PLX3397 treatment. Here, mice with epilepsy were treated with PLX3397 at 3 or 30 mg/kg per day or vehicle alone. Hippocampal mRNA was extracted on day 14 of treatment and module 18 expression was assayed by microarray (see Methods). In keeping with CRAFT prediction, we observed a significant ($P < 2.2\text{x}10^{-16}$) and dose-dependent ($P =$

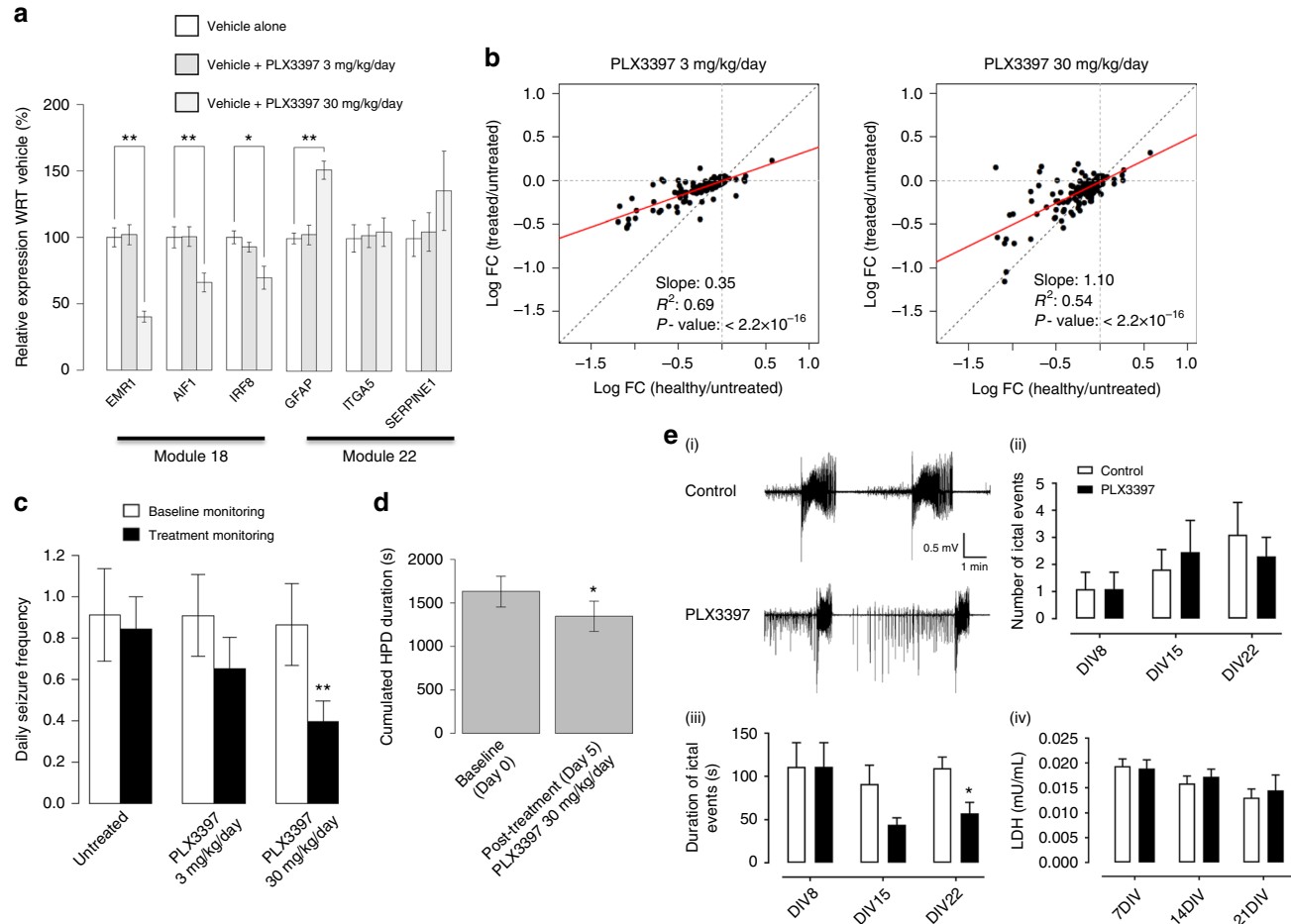

**Fig. 4** Effect of PLX3397 on module 18 expression and seizures. **a** PLX3397 regulates module 18 genes. Epileptic mice were treated daily for 7 days with vehicle or PLX3397 at 3 or 30 mg/kg per day ($n = 8$ mice in each group). At the end of the treatment, hippocampal RNA was extracted and the expression of marker genes in modules 18 and 22 was assayed by rt-qPCR. Module 18 marker genes were significantly down-regulated by PLX3397 at 30 mg/kg per day (*$P < 0.05$, **$P < 0.01$—one-tailed Welch's $t$ test). **b** Restoration of module 18 expression in epilepsy toward health by PLX3397. Epileptic mice were treated with PLX3397 at 3 or 30 mg/kg per day or vehicle alone ($n = 8$ mice in each group). Hippocampal mRNA was extracted on day 14 of treatment and module 18 expression assayed by microarray. Red line indicates the linear negative correlation between the two conditions compared to a theoretical complete restoration of expression toward the healthy state (dotted black line). Treatment with PLX3397 resulted in a significant and dose-dependent ($P = 3.6 \times 10^{-14}$) restoration of expression of module 18 toward health (i.e., toward the dotted black diagonal). **c** Efficacy of PLX3397 on seizures—pilocarpine model. Epileptic mice were baseline monitored for a week (white) before daily administration with vehicle or PLX3397 at 3 or 30 mg/kg per day ($n = 20$ mice in each group) and monitored for a second week (black). PLX3397 treatment induced a significant decrease in daily seizure frequency (**$P < 0.01$—Wilcoxon's signed-rank test) at 30 mg/kg per day. **d** Efficacy of PLX3397 on paroxysmal hippocampal discharges—kainate model. Epileptic mice ($n = 8$) were EEG monitored at baseline (day 0) for 2 h prior to daily administration of PLX3397 at 30 mg/kg per day for 4 days and then EEG monitored on day 5 for 2 h to assess drug efficacy. Treatment with PLX3397 led to a significant (*$P < 0.05$) reduction in the duration of HPDs. **e** Efficacy of PLX3397 in organotypic hippocampal slice cultures. (i) Representative field potential traces of ictal epileptiform activity in dentate gyrus (DG) granular cell layer following 2 weeks of vehicle alone or 1 µM PLX3397, (ii) mean and (iii) duration of ictal events (±S.E.M.) at baseline (DIV 8) and following PLX3397 (DIV 15 and DIV 22). (iv) supernatant concentrations (mean ± S.E.M.) of lactate dehydrogenase at baseline (DIV 7) and following PLX3397 (DIV 14 and DIV 21). In total, 60 hippocampal slices from six rats were analyzed consisting of 36 slices for control and 24 for PLX3397 treatment groups, respectively. *$P < 0.05$

$3.6 \times 10^{-14}$) restoration of module 18 expression toward health following treatment with PLX3397 (Fig. 4b).

These results are consistent with a shift in the expression of module 18 in epilepsy toward the healthy state following PLX3397 exposure. However, module 18 is predicted to be highly expressed in microglia (Supplementary Figure 2b), and it has been suggested that PLX3397 may deplete the brain microglial cell population as judged by Iba1 immunolabeling[33]. Using epileptic mouse cortex and hippocampus samples following treatment with PLX3397, we observed a similar decrease in Iba1 immunolabeling (Supplementary Figure 8). However, Iba1 (encoded by *Aif1*) is both a component of module 18 *and* a

predicted regulatory target of Csf1R and therefore Iba1 staining alone cannot distinguish between depletion of microglia cells by PLX3397 or a more focused down-regulation in *Aif1* gene expression. To distinguish between these two possibilities we undertook a series of further analyses.

First, since module 18 is only one of three microglial modules (the others being modules 16 and 24), if the measured change in expression of module 18 following PLX3397 treatment is a consequence of microglia cell loss rather than a focused transcriptional change, then all three modules should be enriched for genes down-regulated by PLX3397. Of these three modules, however, only module 18 was enriched (FDR <0.01) for genes

down-regulated by PLX3397 (Supplementary Figure 9), suggesting a selective effect on module 18 expression by PLX3397 as opposed to a global down-regulation of microglial modules due to gross microglial depletion. Consistent with this interpretation, we observed that the expression of the microglia-specific gene Sall1[35] was up-regulated by PLX3397 (Log$_2$ fold change = 0.17, FDR = $2.16 \times 10^{-6}$).

We then investigated the effect of PLX3397 on mouse primary microglia cells in vitro (see Methods). Supplementary Figure 10 shows that treatment of both activated and basal microglia with 1 µM PLX3397 is not associated with detectable microglial cell death as assayed by immunocytofluorescence.

We then assessed the effect of PLX3397 on the microglial transcriptome in primary microglia cells, again assessing the effect of PLX3397 in basal and activated states. For these experiments, we generated genome-wide RNA-seq profiles for each of the following four conditions (1) basal microglia, untreated, (2) basal microglia, PLX3397 treated, (3) activated microglia, untreated, and (4) activated microglia, PLX3397 treated. The full list of microglia genes differentially expressed following treatment with PLX3397 in basal and activated microglia are reported in Supplementary Data 12 and 13, respectively. Using the list of genes differentially expressed in microglia by PLX3397, we investigated whether PLX3397 could induce pathways for programmed cell death. We curated gene lists for all known pathways related to apoptosis from GO, KEGG, Panther, Reactome, and Wiki pathways (total number of apoptosis gene sets = 81), and tested whether any of these pathways were induced by PLX3397 using gene set enrichment analysis. Of the 81 pathways, none were significantly (FDR <0.05) induced by PLX3397 in microglia in either basal (Supplementary Data 14) or activated (Supplementary Data 15) states. In contrast, we observed that module 18 is highly significantly down-regulated by PLX3397 in primary microglia in both basal ($P = <1.0 \times 10^{-5}$) and activated ($P < 1.0 \times 10^{-5}$) states (Supplementary Data 16).

Taken together, these analyses point to a PLX3397-dependent reversion of module 18 expression in epilepsy toward health via Csf1R inhibition. Under the signature reversion paradigm, if module 18 has been correctly assigned as a driver of epileptic seizures then PLX3397 treatment should exert a therapeutic effect in epilepsy. We decided to test this prediction by investigating the therapeutic effect of PLX3397 in pre-clinical models of epilepsy.

**Pre-clinical assessment of Csf1R inhibition.** We first assessed PLX3397 efficacy in epilepsy using the same pilocarpine model of TLE used to generate the gene expression data for our co-expression analyses (above). Epileptic mice underwent 14 days of continuous motion sensing 3D accelerometry synchronized with continuous video monitoring to determine their baseline seizure frequency. This was followed by daily treatment with PLX3397 (3 or 30 mg/kg per day) or vehicle alone (20 mice in each group) for 14 days with continuous monitoring of seizures during the treatment phase. Treatment of epileptic mice with PLX3397 30 mg/kg per day resulted in a significant ($P < 0.01$) decrease seizure frequency (Fig. 4c). Mortality was recorded to ensure no differences in the severity of SE between the groups or treatment effects on survival: one mouse died in the 3 mg/kg per day arm, one mouse died in the 30 mg/kg per day arm, and no lethality was recorded in the pilocarpine/vehicle arm.

Next, to confirm the therapeutic effect of PLX3397 on epileptic seizures, we repeated the efficacy analysis using the mouse intrahippocampal kainate model of TLE (see Methods) [36]. Here, seizure activity was assessed using electroencephalographic (EEG) recordings[23]. The primary clinical outcome was

reduction in the duration of hippocampal paroxysmal discharges (HPDs) in response to PLX3397 treatment, which is a standard efficacy outcome measure in this epilepsy model. In the kainate model, treatment with standard AEDs usually only achieves a reduction in HPD duration at supra-therapeutic doses, suggesting the kainate model is a model of drug-resistant epilepsy[37]. Epileptic mice were EEG monitored at baseline (day 0) for 2 h prior to daily administration of PLX3397 at 30 mg/kg per day for 4 days and then EEG monitored again on day 5 for 2 h to assess drug efficacy. Treatment with PLX3397 was associated with a significant ($P < 0.05$) reduction in the duration of HPDs (Fig. 4d).

Finally, to provide further evidence for the anti-seizure effect of PLX3397, we then assessed PLX3397 efficacy using the ex vivo organotypic hippocampal slice culture (OHSC) model of epilepsy[38]. Whilst pre-clinical testing in animal models remains the mainstay for determining efficacy of candidate antiepilepsy drugs[19], ex vivo OHSCs retain many of the key phenotypic features of acquired epilepsy including a latent period prior to the occurrence of spontaneous ictal events[39], and the developmental sequence of interictal spikes to spontaneous ictal events in OHSCs closely mimics the temporal progression observed in the kainate model of epilepsy[40]. The ex vivo OHSC model of epilepsy therefore allows a multidimensional assessment of PLX3397 efficacy and mechanism beyond the in vivo models considered above. For example, because OHSCs represent tissue isolated from the systemic blood supply or wider brain, the development of epilepsy in OHSC's reflects the intrinsic properties of the hippocampal slice isolated from potential infiltration of exogenous inflammatory (or other) cell types. In Fig. 4e we report the assessment of PLX3397 on epileptiform activity in OHSCs using multi-electrode arrays (MEAs) (see Methods for experimental details). As with the kainate mouse model of epilepsy, treatment with PLX3397 led to a significant ($P < 0.05$) reduction in the duration of ictal events. Concurrent assessment of lactate dehydrogenase (LDH) in the OHSC culture supernatants at baseline before PLX3397 exposure and after the first and second week of PLX3397 treatment revealed no evidence that the anti-seizure effect of PLX3397 was dependent on microglial (or other) cell death (Fig. 4e).

To assess the specificity of Csf1R blockade by PLX3397, we investigated the pharmacokinetics (PKs) of PLX3397 in mice (see Methods). First, we wanted to confirm sufficient duration of exposure of PLX3397 following oral gavage. Following treatment of epileptic mice (pilocarpine model) at 3 and 30 mg/kg per day (per os (p.o.)) for 1 week (8 mice in each group), we analyzed PLX3397 levels in the plasma and brain (the latter 24 h after the last oral administration). Supplementary Figure 11A shows that PLX3397 concentrations were stable for the 24 h period following last oral administration of a 30 mg/kg dose (the therapeutic dose in this animal model of epilepsy) with a free (i.e., protein unbound active) concentration in the mouse brain of approximately 1 nM. Mean free plasma concentrations measured 24 h after the last administration were $0.6 \pm 0.2$ and $13 \pm 2$ nM for 3 and 30 mg/kg doses, respectively, and mean free brain concentrations were $0.06 \pm 0.02$ and $1.0 \pm 0.13$ nM, respectively (Supplementary Figure 11B). Our results confirm that approximately 5% of PLX3397 enters the brain after oral administration as previously reported[33]. In cellular assays of PLX3397 selectivity[41], the half maximal inhibitory concentration (IC$_{50}$) of PLX3397 for inhibiting Csf1R is 20 nM, compared to IC$_{50s}$ for c-Kit and Flt3 of 120 nM and 1.7 µM, respectively (Supplementary Figure 11B) indicating that free brain concentrations of PLX3397 attained after 30 mg/kg per day oral gavage are within a range for Csf1R kinase activity, but substantially below that of either c-Kit or Flt3.

Taken together, these drug efficacy, cell viability, and gene expression analyses are consistent with microglial Csf1R inhibition exerting a therapeutic effect in acquired epilepsy in the absence of microglial depletion. We therefore evaluated whether PLX3397 has a detectable effect on the microglial phenotype (see Methods).

We first assessed the microglia phenotype using ex vivo brain slices from epileptic mice (pilocarpine model) treated with vehicle or PLX3397 at 30 mg/kg (the therapeutic dosage in epileptic mice). Analysis of phagocytic function assessed by pH-sensitive rhodamine-labeled zymosan particle uptake revealed no significant differences in the number of particles taken up by microglia in the brain slices of vehicle or PLX3397-treated mice (Supplementary Figure 12A). In keeping with the evidence from ex vivo hippocampal slices and in vitro primary microglia cell viability assays (above), we found no evidence that PLX3397 impacts microglia cell viability as measured by LDH activity in the brain slice supernatant (Supplementary Figure 12A). We then assessed microglia morphology in brain slices from vehicle and PLX3397-treated epileptic mice and identified that microglia in epileptic mice brains exposed to PLX3397 have thicker healthier processes compared to the highly filamentous discontinued filopodia in epileptic mice treated with vehicle alone (Supplementary Figure 12B). Finally, using mouse primary microglia, we assessed the effect of PLX3397 on microglial cell migration using the in vitro scratch assay. At 24 h we observed a significant decrease in the motility of microglia exposed to 1 µM PLX3397 (Supplementary Figure 12C) and again confirmed the viability of microglia cells at 1 µM PLX3397 concentration (Supplementary Figure 12D).

Overall, these data point to PLX3397 having a disease context-specific effect on epilepsy via module 18 as predicted by CRAFT. Therefore, to provide further evidence for PLX3397's context-specific effect on epilepsy (i.e., consistent with its measured effect on module 18 gene expression and the microglial phenotype), we investigated whether PLX3397 has anti-seizure effects in normal (non-epileptic) mice induced to have seizures (see Methods). For these studies, we used three different acute seizure models, the maximal electroshock seizure (MES) model, the electrical 6 Hz psychomotor model and the chemical pentylenetetrazol (PTZ) model. In Supplementary Figure 13, we show that neither single dose PLX3397 nor chronic pre-treatment with PLX3397 has any anti-seizure effects in any of the acute seizure models across a broad range of outcome measures. These data clearly distinguish PLX3397 from all standard AEDs, which are effective in at least one of these acute seizure models, and indeed, the MES, 6 Hz, and PTZ acute seizure models have been the traditional gatekeepers for new AED discovery for over half a century[42].

## Discussion
In this study, we used a gene network perspective of disease as a landscape for drug discovery. Under this framework, restoration of disease-related module expression toward health is considered to be predictive of therapeutic benefit, allowing "target" validation at the earliest stage of the drug discovery process. Based on this premise, we set out to develop and validate a predictive gene regulatory framework for target discovery. Given the tractability of cell membrane receptors as drug targets, and the large number of drugs that already target cell surface receptors, we aimed to connect module expression to cell membrane receptors.

Starting from genome-wide gene expression profiling of the epileptic mouse hippocampus, we first identified co-expression networks (modules) associated with the epileptic condition. The cell-type specificity of these modules and their functional processes was assessed using enrichment analyses, and the regulatory influence of cell membrane receptors over the selected modules was then inferred using gene regulatory information in a causal reasoning framework (CRAFT).

Of the cell surface receptors predicted by CRAFT to influence the expression one or more candidate epilepsy module, we chose to validate Csf1R because of an absence of prior information connecting Csf1R to epilepsy and the availability of a tool compound (PLX3397) by which to test CRAFT's predictions related to Csf1R's regulation of module 18 (and by extension module 18's relationship to epilepsy). Analysis of module 18 expression in the epileptic mouse brain revealed a PLX3397-dependent restoration of module expression toward health. The predicted therapeutic effect of PLX3397 on epilepsy was then confirmed in three independent models of epilepsy, including a mouse model of pharmacoresistant epilepsy—a model in which traditional AEDs at standard doses are ineffective. In addition to validating CRAFT as a predictive framework for drug target discovery, these results identify Csf1R inhibition as a potential novel therapeutic strategy in epilepsy and provide further evidence to support the role of innate immunity in the occurrence and maintenance of seizures in acquired epilepsy[14,43].

Csf1R is a membrane receptor expressed by myeloid lineage cells including monocytes, macrophages, and microglia[44]. It has been suggested that microglia are dependent on Csf1R signaling for their survival such that brain microglia are reported to be depleted from naïve mice following prolonged high dose treatment with PLX3397 (5–6 times higher exposure than in our study)[33]. In our study, we found no evidence for microglial depletion by Csf1R in a series of ex vivo and in vitro experiments and further we identified that the microglial marker Iba1 (encoded by *Aif1*) is a predicted target of Csf1R and expected to be down-regulated by Csf1R blockade. These results highlight the dangers of interpreting reduced Iba1 expression as microglial cell depletion. In our study, we present multiple layers of evidence through genomic and microglial functional readouts to suggest that PLX3397 has an effect on module 18 expression and microglial phenotpye in the absence of microglial cell death.

The ability to map the landscape of a disease in terms of its gene regulatory relationships offers considerable opportunities to accelerate the drug discovery process. Although we took advantage of experimentally validated interactions between TFs and target genes and between membrane receptors and TFs, meta-databases such as MetaBase® have limitations in terms of the accuracy and completeness of this information, which places restrictions on the scope and accuracy of our target predictions. For example, the direction of effect of an interaction is often not specified in a database, and the relationship between membrane receptors and TFs is currently determined using linear pathways where knowledge is still incomplete. However, as the completeness of our knowledge of these regulatory relationships improves, including more detailed knowledge of cell-type-specific interactions between TFs and target genes, so the accuracy and scope of CRAFT is also expected to improve. At present, the major challenge was to establish proof of concept that gene regulatory knowledge combined with causal reasoning offers a valid framework for discovering mechanistically novel membrane receptors as drug targets, and this is what the framework described here makes possible. Although our causal reasoning framework was implemented using regulatory interactions from Clarivate Analytics MetaBase®, several other databases provide similar sources of information that can be adapted to the CRAFT framework. For example, biological pathway databases such as the Reactome pathway Knowledgebase[45] and Pathway Commons[46] provide well-characterized linear signaling pathways which can be used to connect membrane receptors to TFs, whilst TF target databases such as TRRUST[47] provide information relating TFs to

target genes. As well as having utility in prioritizing novel drug targets for disease, CRAFT's causal reasoning framework may ultimately have broader biological value in terms of understanding and modulating maladaptive transcriptional responses to environmental perturbations.

From a clinical perspective, our study identifies Csf1R as a novel drug target for acquired epilepsy, and highlights and further supports the use of immunomodulatory therapies as a valid therapeutic approach in acquired epilepsy[48].

For this study, we chose to develop CRAFT using an established mouse model of acquired epilepsy. This allowed for a standardization of epilepsy cases and controls not possible with human samples due to the substantial batch differences between surgically resected hippocampi from living patients and control post-mortem samples ascertained after death.

Unlike traditional gene expression analyses where power is most often considered in terms of power to detect differentially expressed genes (DEGs), for our study, power related to the sample size required to detect differentially co-expressed modules. To investigate the effect of sample size on our ability to detect modules which are differentially co-expressed between the epileptic and control mouse hippocampus, we implemented a post hoc permutation-based framework that preserves the real data structure and employs random sub-sampling to determine the minimum sample size necessary to identify differentially co-expressed modules. Our simulations (Supplementary Figure 14) revealed that some modules (e.g., module 18) can be detected with as few as 20 epilepsy case and control samples, whilst for others (e.g., module 12) differential co-expression between epilepsy and control status can only be detected when the sample size is much larger ($n = 100$).

In conclusion, CRAFT provides a gene regulatory and causal reasoning framework to identify membrane receptors as novel drug targets from gene expression data. As well revealing Csf1R as a mechanistically novel target for acquired epilepsy, CRAFT highlighted many other candidate regulators of epileptic networks that may warrant further investigation as potential novel anti-epilepsy drug targets. We therefore make our causal reasoning framework in epilepsy and all its results available to the epilepsy scientific community for unrestricted interrogation via a web interface (http://ec2-54-191-145-199.us-west-2.compute. amazonaws.com:3000/#/).

## Methods

**Mouse pilocarpine model of epilepsy.** SE was induced in male Crl:NMRI(Han)-FR mice (each mouse weighing 28–32 g at the beginning of the study) by a single injection of pilocarpine as previously described[22,49]. Briefly, animals were injected intraperitoneally (i.p.) with 1 mg/kg of $N$-methylscopolamine bromide 30 min prior to pilocarpine treatment (300 mg/kg; i.p.). Ten to forty-five minutes after pilocarpine injection the animals displayed generalized clonic–tonic seizures that progressed to continuous convulsive activity, that is, SE. The SE was allowed to persist for 3 h and was then interrupted by i.p. injection of diazepam (10 mg/kg). Mice surviving SE typically show spontaneous recurrent seizures (i.e., epilepsy) within days to weeks and continue to have spontaneous seizures for several weeks[22,49]. All mice underwent continuous monitoring for seizures for 14 consecutive days beginning 28 days following SE prior to sampling the hippocampus and extraction of RNA. Seizure monitoring was performed with a proprietary system (UCB Pharma) using simultaneous recording of locomotor activity with 3D accelerometer and video. This system allows for automated detection of behavioral seizures by analysis of the accelerometry signal, which is then reviewed manually using the time-locked video recordings. All behavioral seizures were scored according to the Racine's[50] method after careful review of corresponding video clips by experienced technical personnel. Only secondary generalized seizures Racine's score 3–5 were quantified and used to calculate total seizure counts. Data from 100 epileptic and age-matched and gender-matched control were included in the study. All in vivo experiments were performed according to the National Rules on Animal Experiments in Belgium and to the guidelines of the European Community Council Directive 2010/63/EU. Analyses were conducted under Imperial College Research Ethics Committee approval ICREC_14_2_11. All efforts were made to minimize animal suffering.

**Post hoc sample size calculation.** Unlike traditional gene expression analyses where power is most often considered in terms of power to detect DEGs, for our study, power relates to the samples size required to detect *differentially co-expressed modules*. To investigate the effect of sample size on our ability to detect modules which are differentially co-expressed between the epileptic and control mouse hippocampus, we implemented a post hoc permutation-based framework that preserves the real data structure and employs random sub-sampling to determine the minimum sample size necessary to detect significantly differential co-expressed modules. This framework was implemented in three steps: (i) sample size was randomly reduced in steps of 10% (i.e., 90, 80, 70, 60, 50, 40, 30, 20, and 10% of the samples), (ii) the empirical significance of differential co-expression was calculated at each sample size, (iii) steps (i–ii) were performed 50 times to assess sampling variation. We then selected three representative differential co-expression modules associated with epilepsy in our study (M12, M18, and M21), which were each of similar size in terms of the number of genes but which varied in terms of the mean gene–gene correlation of each module. Using this permutation-based framework, for each module, differential co-expression was assessed in terms of the empirical significance ($P$ value) of the difference between the mean correlation of module genes in epileptic versus control mice.

Our simulations revealed that as the sample size decreases there is an increase in noise (measured as variation between each bootstrap permutation—the interquartile range in the boxplots in Supplementary Figure 14) for both the observed module correlations and the null distribution from the permutation. As expected, for each module, there is an inverse relationship between effect size (i.e., the difference in mean correlation of module genes between conditions) and sample size. Since mean gene–gene correlation varies between modules, the differential co-expression of some modules (e.g., M18) can be detected with as few as 20 mouse samples, while for others (M12 for example), the differential co-expression between epilepsy and control status can only be detected when the sample size is much larger—in the case of M12 only when sample size is $n = 100$. These results show that some epilepsy modules would not have been reliably detected as differentially co-expressed with a sample size less than $n = 100$.

**Sample preparation for RNA-seq analysis.** Total RNA was extracted from the left hippocampus of each mouse ($n = 200$; 100 case and control mice). Sample preparation for RNA-seq was performed according to the protocols recommended by the manufacturers (TruSeq RNA Kit, Illumina). Sequencing was done using Illumina HiSeq 2000 sequencer, with paired-end 75 bp nucleotide reads according to the protocol recommended by the vendor. Raw reads were mapped to the reference mouse genome (mm10) using TopHat version 2.0.8[51]. Reads were annotated using "union" gene model from HTSeq package version 0.6.

**DE analysis.** Genes were considered "expressed" and included in the analysis if they had an expression value of $\text{Log}_2$ FPKM $>0$ in at least 5% of the samples across cases and controls. DE analysis was performed using the Bioconductor package EdgeR, which implements generalized linear model based on negative binomial test model for RNA-Seq in R[52]. $P$ values were corrected for multiple testing using BH FDR[53]. A cut-off of FDR ≤0.05 was applied to select DEGs.

**Co-expression network analysis.** Co-expression networks were constructed using hierarchical clustering of normalized gene expression profiles from 100 epileptic mice hippocampi. First, for all genes expressed in the hippocampus, we calculated 1-Spearman's correlation coefficients (called Spearman's distance) as a distance metric between the expression of any two genes[54–56]. Second, the distances between any gene pair were partitioned (clustered) using the Ward's clustering method[57] and organized into a dendrogram. Briefly, starting from the matrix of Spearman distances, Ward's method uses a recursive clustering procedure to form partitions (clusters) of genes. At each step of the procedure, the Ward clustering minimizes the loss of information (measured as the error of sum of squares in the Spearman distances) associated with each grouping of genes. To identify discrete clusters, we recursively cut the dendrogram to generate 299 clustering configurations that included from $K = 2$ to $K = 300$ clusters. In order to identify the optimal and stable number of clusters ($K_x$), we calculated the percentage of the variance explained ($R^2$) by each clustering configuration (i.e., $R^2$ for each considered $K$) as follows:

$$\text{Percentage of variance explained}\left(R^2\right) = \frac{\text{BSS}}{\text{WSS} + \text{BSS}}$$

where BSS (between sum of squares) is the between-groups variance in Spearman distances and WSS (within sum of squares) is the within-groups variance in Spearman distances. We used two criteria to choose the value $K_x$ for which the variance explained reaches a plateau, that is, there is no additional gain in information ($R^2$) when using the next clustering $K_x + 1$ (Figure S1). The criteria used to choose $K_x$ were (1) the "Elbow" (or "knee") method[25] and (2) the pseudo F-index[58]. Both criteria indicated an optimal and stable number of clusters, $K_x = 29$ (Figure S1).

**Differential co-expression analysis.** For each cluster the correlation between gene expression profiles was computed in healthy and in epileptic animals

separately and the difference in co-expression measure was based on Euclidian distance between the two distributions (i.e., between healthy and epileptic mice). The statistical significance for the difference in co-expression was assessed according to the null distribution generated by performing 10,000 permutations of genes in the module[27]. This empirical $P$ value of significance was estimated for each cluster and then corrected for the number of clusters tested for differential co-expression using BH correction (BH-adjusted). A cluster was considered to be significantly differentially co-expressed if its BH-adjusted empirical $P$ value was <0.05.

**Conservation in human TLE**. Using the lists of genes from the co-expression clusters in the mouse epileptic hippocampus, we ascertained "one2one" human orthologs from the biomart Ensembl database. For epileptic cases, we used genome-wide expression in PubMed whole human hippocampus samples from 122 epilepsy patients who had undergone selective amydalohippocampectomy for mesial temporal epilepsy (mTLE) with hippocampus sclerosis (HS), downloaded from GSE63808[14]. For non-epileptic controls ($n = 63$), we used genome-wide gene expression data generated from 63 human hippocampus samples with no history of neurological or psychiatric disease, downloaded from GSE45642[31]. The differential co-expression test was performed as described above.

**Association of co-expression modules with total number of seizures**. First, the frequency of behavioral seizures in each epileptic mouse was recorded by 14 days of continuous motion sensing 3D accelerometry synchronized with continuous video monitoring starting on day 28 post SE (as described above). Next, the relationship between module expression and seizures was explored in two ways. First, we explored the correlation between each module's eigengene (i.e., its PC1) and seizure frequency using Spearman's correlation. Second, the relationship between the expression of a module and seizures we explored by first calculating the Spearman's correlation of expression of each individual gene in a module with seizure frequency (termed QTT analysis[29]), and then tested the enrichment of each module for genes individually correlated with seizure frequency (at FDR <0.05).

**Cell-type enrichment analysis**. Cell-type enrichment analysis was performed for nine major cell types (cortical pyramidal neurons, CA1 pyramidal neurons, interneurons, astrocytes, endothelial cells, mural cells, oligodendrocytes, ependymal cells, and microglia) using marker gene signatures obtained from single-cell RNA-seq of the mouse hippocampus[26] (Fisher's exact test). BH correction for multiple testing was done and significance threshold was at FDR ≤0.05.

**Literature analysis to identify modules enriched in epilepsy-associated genes**. To extract epilepsy-associated genes from the scientific literature, gene–epilepsy co-citation was searched in PubMed abstracts (3,811,179 abstracts, with at least one gene–literature pair) using SCAIview webserver as available on 28 May 2014. The weight of evidence relating a particular gene to epilepsy was then quantified by determining if that gene's co-citation with epilepsy (23,092 Abstracts with at least one gene–epilepsy co-citation) was more frequent than expected by chance (hypergeometric test). The list of epilepsy-associated genes was established by applying a threshold of FDR ≤0.05. Then, by considering gene–epilepsy pairs significant at FDR <0.05, the modules were ranked according to their enrichment of gene–epilepsy pairs (hypergeometric test).

**In silico causal reasoning (CRAFT framework)**. To predict drug targets, we aimed to connect disease-associated co-expression modules to regulatory TFs that in turn can be modulated by membrane receptors. To implement this analysis, we designed a causal reasoning framework that takes into account the direction of effects between the three components of the system, that is, membrane receptor > TF > target genes. The interactions between these three components and the direction of these interactions were obtained from the Clarivate Analytics Meta-Base® (version 6.15.62452, https://clarivate.com/products/metacore/), which is a meta-database of manually curated literature-based contextual biological interactions. Prior to the causal reasoning analysis, we removed all membrane receptors and TFs that were not expressed in the mouse hippocampus, resulting in a list of 1624 TFs and 307 receptors, and we also considered only those target genes that were expressed in the mouse hippocampus. Next, for each TF, we identified the set of genes targeted by that TF and considered the activity of the TF on the target genes in one or more of three possible categories: (a) genes activated by the TF (act), (b) genes inhibited by the TF (inh), and (c) genes reported to be regulated by the TF in MetaBase® but where the direction of the interaction was unknown (unk). Next, the effect of each receptor on a TF was assessed using MetaBase® defined canonical linear pathways, with the direction of the effect of the membrane receptor on the TF considered as either activating (act), inhibitory (inh), or unknown (unk). Note, hereon the term "regulator" can refer to either a TF or a membrane receptor. The transcriptional effect of a regulator is defined by the set of target genes whose expression is predicted to be influenced by that regulator. For regulators that are membrane receptors, the directionality of the receptors on network expression was defined by the causal reasoning rules set out in Table 1, below.

**Table 1 Causal reasoning rules for membrane receptor effects on target gene expression**

| Receptor | Transcription factor | Target genes |
|---|---|---|
| Act (+) | Act (+) | Activated |
| Act (+) | Inh (−) | Inhibited |
| Inh (−) | Act (+) | Activated |
| Inh (−) | Inh (−) | Activated |

Next, to assign directionality to the activity of a regulator in terms of disease context, we considered the effect of the regulator on gene expression in the context of whether the target genes in the co-expression module were over-expressed ("o") or under-expressed ("u") in epilepsy. The set of genes in a module over-expressed or under-expressed in epilepsy were termed "sub-modules" (of the parental module). The significance of effect of a regulator on a sub-module was then assessed by testing the overlap between genes under the control of the regulator (i.e., a membrane receptor or a TF) and the genes belonging to a sub-module (hypergeometric test), taking all genes under the control of the regulator as the universe. We report the effect of membrane receptors and TFs on sub-module gene expression separately in Supplementary Data 10 and 11. FDR was calculated using BH correction of enrichment $P$ values, taking into account the total number of enrichment tests performed. Regulators were considered to exert a significant effect on sub-module expression at FDR <0.05 and reported graphically in Supplementary Figure 7. Because receptors may act via a number of different TFs, and because TFs themselves may influence the expression of many genes, sometimes in opposing directions, then for a regulator to be considered an activator (act) of a sub-module, the regulator needed to exert a significant (FDR <0.05) positive effect on the expression of a sub-module but have no significant (i.e., FDR >0.05) negative effect on the sub-module's expression. Similarly, for a regulator to be considered an inhibitor (inh) the regulator should have a significant negative effect on sub-module's expression (FDR <0.05) and no significant positive effect (FDR >0.05). Regulators with significant (FDR <0.05) positive and negative effects on sub-module expression were deemed to have an unspecified (uns) effect on sub-module expression.

The impact of a regulator on sub-module expression was then quantified according to the following two statistics:

1. The proportion of genes regulated by the "regulator" which are in the sub-module = activity.
2. The proportion of genes in the sub-module that are under the control of the regulator = weight.

Using these two values, regulators were ranked as follows:

(A) Absolute ranking: Regulators with a defined impact, that is, "act" or "inh", were considered to have a higher priority over regulators with an unspecified effect. The regulators were then sorted according to the sum of the ranks of their activity and weight.

(B) Relative ranking: Because modules may be regulated by more than one regulator, and because the target genes in a sub-module for each regulator may overlap, where more than one regulator was identified, for each we re-calculated the absolute rank based on the target genes not shared with the previous regulator (based on the absolute rank list) in order to maximize sub-module coverage and to identify potential synergistic combinations of regulators. The first regulator is the one with the highest absolute rank.

**Pre-clinical evaluation of Csf1R blockade in mice**. To evaluate the effect of Csf1R blockade on gene network expression and epilepsy, we first used a separate cohort of chronic epileptic NMRI mice (pilocarpine model as above). All mice underwent screening consisting of 7 continuous days (starting 6–7 weeks after pilocarpine-induced SE) to confirm the presence of spontaneous recurrent seizures (i.e., epilepsy). During the eighth post SE week, $n = 64$ chronic epileptic mice (Racine's score 3–5) were video monitored (as described above) for a further 14 consecutive days to establish their baseline seizure daily frequency, which was calculated by dividing the total number of seizures by the exact duration of monitoring period. Subsequently, the mice were randomly assigned into three separate groups ($n = 21$–22) to ensure comparable seizure frequency between the groups at baseline. One group then received daily injections (oral gavage) with vehicle, while the two other groups received the Csf1R inhibitor PLX3397 at 3 and 30 mg/kg per day for 14 days. All three groups were continuously monitored and their daily seizure frequency was calculated and compared to baseline. To analyze the effect of PLX3397 treatment on the daily seizure frequency, Wilcoxon's signed-rank test was applied between the baseline and the treatment monitoring periods within each treatment group. Mice were sacrificed for sampling of the brain on the day after the last injection with either vehicle or PLX3397. Since in the pilocarpine model symmetrical and bilateral pathology is observed in the brain, one hemisphere of the brain was processed for immunohistochemistry (see below), while the

hippocampus from the other hemisphere was used for RNA extraction and microarray quantification of mRNA expression.

**Immunohistochemistry, image acquisition, and quantification.** Following brain dissection, right brain hemispheres were immediately fixed by immersion in 4% paraformaldehyde overnight at 4 °C. Samples were rinsed in several changes of cold 1× phosphate-buffered saline (PBS), cryo-protected by overnight immersion in 15% sucrose solution at 4 °C, and then embedded in OCT (Tissue-Tek) and quickly frozen over liquid nitrogen. Hemispheres were serially cut with a cryostat microtome into 12 µm coronal sections that were mounted on slides and stored at –40 °C until treatment. For immunohistochemistry, sections were rinsed twice in PBS, and then incubated in PBS containing 0.3% Triton X-100 and 5% normal goat serum (PBS-T) for 1 h at room temperature to block non-specific binding sites. Anti-Iba1 primary antibody (1:1000; rabbit; Synaptic Systems) was diluted in PBS-T and applied on sections overnight at room temperature in a humidified chamber. Labeling was visualized using Alexa Fluor secondary antibodies 488 (1:400; Invitrogen) and sections were counterstained with 4′,6-diamidino-2-phenylindole (DAPI). Whole slide imaging was performed by fluorescence microscopy using NanoZoomer-XR with ×40 objective (Hamamatsu). Automatic quantification of cell density in specific brain regions was performed using the VisioPharm 5 software (VisioPharm).

**Total RNA extraction from hippocampi.** Snap-frozen hippocampi were homogenized in RLT buffer (Qiagen, 74134) using Precellys® system in CK14 tubes following the manufacturer's instructions. Total RNA was obtained from mouse hippocampi using RNeasy Plus Mini Kit (Qiagen, 74134) following the manufacturer's instructions. RNA concentration was measured with a NanoDrop ND-1000 Spectrophotometer. RNA quality was assessed using the Bio-Rad Experion Automated Electrophoresis System (RQI ≥7).

**Quantitative polymerase chain reaction.** Complementary DNA (cDNA) was synthesized from 1 µg total RNA using Applied Biosystems High-Capacity cDNA Reverse Transcription Kit in a total volume of 100 µl following the manufacturer's protocol. qPCR reactions were performed using a CFX384 Real-Time System. Two microliters of 2× diluted cDNA were analyzed in triplicate for genes of interest (see Supplementary methods) expression using inventoried TaqMan® Gene Expression Assays (ThermoFisher Scientific) and TaqMan® Universal PCR Master Mix (ThermoFisher Scientific) in a final volume of 10 µl according to the manufacturer's recommendations. Cq values were obtained from the Bio-Rad CFX Manager 3.1 software using regression determination mode. Normalized relative expression levels were calculated using qbase + software[59] (Biogazelle NV, Zwijnaarde, Belgium). Among eight genes, *Brap* and *Bcl2l13* were identified with the geNormplus[60] module in qbase+ as the most suitable reference genes and were used for normalization. One-tailed Welch's *t* test was performed on Log 2-transformed normalized expressions.

**Microarray hybridization and analysis.** The quality control (QC), RNA labeling, hybridization, and data extraction were performed at GenomeScan/ServiceXS B.V. (Leiden, The Netherlands). RNA concentration was measured using the Nanodrop ND-1000 spectrophotometer (Nanodrop Technologies, Wilmington, DE, USA). The RNA quality and integrity was determined using Lab-on-Chip analysis on the Agilent 2100 Bioanalyzer (Agilent Technologies Inc., Santa Clara, CA, USA). Biotinylated cRNA was prepared using the Illumina TotalPrep RNA Amplification Kit (Ambion Inc., Austin, TX, USA) according to the manufacturer's specifications with an input of 200 ng total RNA. Per sample, 750 ng of the obtained biotinylated cRNA samples was hybridized onto the Illumina MouseRef-8 v2 (Illumina Inc., San Diego, CA, USA). Hybridization and washing were performed according to the Illumina Manual "Direct Hybridization Assay Guide." Scanning was performed on the Illumina iScan (Illumina Inc., San Diego, CA, USA). Image analysis and extraction of raw expression data was performed with Illumina GenomeStudio v2011.1 Gene Expression software with default settings (no background subtraction and no normalization). QC and normalization of microarray data were performed using "Lumi" package[61] in R-Bioconductor[62]. Briefly, samples that passed QC were Log 2 transformed and normalized by quantiles between chips. DE was calculated between the different groups using "Limma" package[63] in R-Bioconductor.

**Kainate-induced mTLE murine model.** Kainate-induced mTLE model experiments were performed at Synapcell SAS (38700 La Tronche, France). Animal procedures were approved by the ethical committee of the High Technology Animal Platform and performed in accordance with the European Community Council Directive (2010/63/EU). Briefly, C57/BL6 mice underwent surgery under general anesthesia using isoflurane (3% in oxygen). Stereotaxic injection of 50 nl of a kainic acid solution (1 nmol) was performed in the right dorsal hippocampus (AP = −2, ML = −1.5, DV = −2 mm with bregma as reference). After KA injection, mice were implanted with a bipolar electrode in the ipsilateral dorsal hippocampus and a reference electrode over the cerebellum. After surgery, animals were housed in individual cages with access to food and water ad libitum under a 12/12 h light and dark cycle. Eight mTLE mice underwent 2 h baseline EEG recording 4 weeks after KA injection. Chronic treatment started after the baseline monitoring (D0),

where animals were injected daily p.o. (10 ml/kg) with 32.6 mg/kg of PLX3397 for four consecutive days. PLX3397 was formulated in 1% (w/v) methylcellulose (400 cps), 0.1% (w/v) Tween-80, and 0.1% (w/v) silicone antifoam 1510 US in water. An efficacy EEG recording was performed for 2 h on the fifth day (D5). EEG recordings were analyzed for identification/quantification of HPDs. Individual data were expressed as mean cumulated duration and mean number of HPDs per h over the 2-h EEG recording period. The effect of the tested compound was compared using one-way analysis of variance (ANOVA) for repeated measures followed by paired comparison versus baseline period (D0).

**Slope test.** Slope test was performed to test the relationship between PLX3397 dose and restoration of network gene expression back to "normal." The test was performed using "smartr" version 3.4.3, R package, with null hypothesis that there is no difference in the slopes obtained from two doses of PLX3397.

**Organotypic hippocampal slice cultures.** All animals used in this study were performed according to the guidelines of the European Community Council Directive 2010/63/EU. All experimental protocols using animals were reviewed and approved by the ethical committee at UCB Biopharma. OHSCs were prepared using polydimethylsiloxane mini-wells as described previously[64]. Briefly, hippocampi were dissected from 6- to 9-day-old Sprague–Dawley rats and cut into 350-µm-thick slices using a McIlwain tissue chopper. Slices were kept in ice-cold Gey's balanced salt solution containing 0.6% D-glucose and 300 µM kynurenic acid (all from Sigma-Aldrich) for 30 min. After recovery, slices were washed three times with the culture medium containing Neurobasal A, B27 supplement, 0.5 mM Glutamax (all from ThermoFisher Scientific), and 30 µg/ml gentamicin (Sigma-Aldrich). Only slices from the middle part of the hippocampus were selected and placed in the center of the mini-wells, which allowed a good positioning over the MEA assembly (200/30, 60 ITO from Multi Channel systems). For immunostaining experiments, slices were cultured on glass coverslips containing the mini-wells in a 6-well culture plate. OHSCs were maintained in a humidified $CO_2$ incubator at 35 °C and the medium was refreshed 2–3 times a week.

**MEA recording and data analysis.** An assessment of spontaneous epileptiform activity was performed as described previously[64]. Field potential recordings were performed once per slice at days in vitro (DIV) 8 (baseline before PLX3397 exposure) and then on DIV 15 and DIV 22, that is, after first and second week of exposure to 1 µM of PLX3397. Extracellular field potential recording and stimulation were performed using the MEA2100 system (2 × 60 channels, Multi Channel Systems). OHSCs grown on MEAs at different DIVs were quickly moved to the MEA recording chamber and spontaneous activity was recorded for 40 min using the MC Rack software (Multi Channel Systems). The recording pads were preheated at 35 °C and 95% O2/5% $CO_2$ gas was continuously flowing into the chamber while recording. At the end of the recording, OHSCs showing no activity or only interictal activity were further examined for viability and synaptic response. Extracellular field potentials were evoked in the CA1, CA3, and dentate gyrus (DG) regions followed by biphasic current pulse stimulation (±30 µA) to Schaffer collaterals, stratum lucidum, and perforant path, respectively. Only slices showing >500 µV of peak-to-peak field excitatory postsynaptic potential amplitude in each region were selected for data analysis. In the study, 9 of 60 slices showing only interictal-like activity did not meet the QC criteria described above and therefore were not included in the data analysis. Data were collected at 1 kHz sampling frequency. Since epileptiform activity is highly synchronized in all subregions of the hippocampus[64], we chose one electrode in the DG granular cell layer to analyze epileptic activity. Since cultured slices were required to adjust to the recording chamber, we excluded signals from first 10 min and only last 30 min of recorded data were used for the data analysis. Automatic detection of ictal events was performed using Labview (National Instruments). Interictal epileptiform discharges were defined as paroxysmal discharges that are clearly distinguished from background activity, with an abrupt change in polarity occurring at low frequency (<2 Hz). Ictal epileptiform discharges were defined as paroxysmal discharges lasting more than 10 s occurring at higher frequency (≥2 Hz). If the next ictal event occurs within 10 s after the previous one, we considered the two as one ictal event.

**LDH assay.** Culture supernatants were collected on DIV 7 (baseline before PLX3397 exposure) and then on DIV 14 and DIV 21, that is, after first and second week of exposure to 1 µM of PLX3397. Immediately after collection, the culture supernatants were kept at −80 °C until the assay was performed. Experiment was performed according to the manufacturer's protocol. LDH was measured as a maker of cell death[65] using the LDH Assay Kit (BioVision). The culture supernatants were diluted 2–3-folds with the assay buffer (BioVision) and colorimetric signals were measured at 450 nm by a microplate reader FlexStation 3 (Molecular Devices). LDH concentrations were calculated from the level of LDH activity that converts nicotinamide adenine dinucleotide (NAD) to NAD reduced (NADH). The concentration was expressed as mU/ml, where one unit of LDH generates 1 µM NADH per minute. Statistical differences were assessed using the GraphPad Prism v.7 software (GraphPad Software Inc., La Jolla, CA, USA) by two-way ANOVA with Fisher's least significant difference post hoc test. $P < 0.05$ was considered significant.

**Assessment of PLX3397 in acute seizure models**. Animals: Male Crl:NMRI (Han)-FR mice weighing 20–30 g were used in all experiments. The animals were kept on a 12/12-h light/dark cycle with lights on at 0600 hours and were housed at a temperature maintained at 20–21 °C and at humidity of about 40%. The mice were housed in groups of 10 per cage ($38 \times 26 \times 14$ cm$^3$). All animals had free access to standard pellet food and water before random assignment to experimental groups consisting of 10 mice each. All in vivo experiments were performed according to the guidelines of the National Rules on Animal Experiments in Belgium, ARRIVE, and the European Community Council Directive 2010/63/EU. All efforts were made to minimize animal suffering.

Drug treatment: PLX3397 (3 and 30 mg/kg) was administered by oral gavage (p.o.) in separate groups of animals ($n = 10$ per group). Mice received either a single drug dose (acute treatment) or daily doses over 5 days (chronic treatment). Seizure tests were performed 4 h after the single dose or the fifth day of treatment, 4 h after the last dose.

MES model: MES was produced by a stimulator (WITT IndustrieElektronik, Berlin, Germany) using a current of 50 mA delivered with a pulse frequency of 50 Hz for 0.2 s through corneal electrodes[66]. A drop of 0.4% oxybuprocaine hydrochloride (Unicaine, Thea, France) was placed on the eyes before electrical stimulation. The mice were observed for 10 s following stimulation and the incidence of tonic hindlimb extension was noted.

PTZ seizure threshold model: PTZ (intravenous) infusion was performed as previously described[66]. Before infusion mice were briefly restrained and a 30-gauge needle attached to a 0.3-m-long polyethylene tubing (PE-10) was inserted into the lateral tail vein. The needle was gently secured to the tail with plastic tape. The tubing was connected to a syringe mounted on infusion pump (Harvard Apparatus, Holliston, MA, USA). PTZ (5 mg/ml) (Sigma, Bornem, Belgium) was infused at the rate of 0.25 ml/min in freely moving mice. The time from the start of the infusion to the onset of myoclonic, clonic, and tonic convulsions was recorded. The threshold doses of PTZ were calculated according to the following formula: threshold dose (mg/kg) = (PTZ concentration (mg/ml) × infusion rate (ml/s) × infusion duration (s) × 1000)/weight of mouse (g)).

6 Hz model: The 6 Hz model was carried out according to a previously described protocol[66]. Briefly, corneal stimulation (44 mA, 0.2-ms-duration monopolar rectangular pulses at 6 Hz for 3 s) was delivered by a constant-current device (ECT Unit 57800; Ugo Basile, Comerio, Italy). A drop of 0.4% oxybuprocaine hydrochloride (Unicaine, Thea, France) was placed on the eyes before electrical stimulation. During the stimulation, mice were manually restrained and released into the observation cage ($38 \times 26 \times 14$ cm$^3$) immediately after the current application. The seizures were often preceded by a brief period (~2–3 s) of intense locomotor agitation (wild running and jumping). The animals then exhibited a "stunned" posture associated with rearing, forelimb automatic movements and clonus, twitching of the vibrissae, and strub tail. At the end of the seizure, animals resumed their normal exploratory behavior. The animal was considered as having a seizure if it did not resume its normal exploratory behavior within 7 s from the stimulation. The duration and severity (Racine's score) of seizures was also noted.

Statistics: Seizure incidence data in the MES and 6 Hz models were compared with Fisher's exact test. Seizure duration and seizure severity data in the 6 Hz model, as well as the threshold doses of PTZ for induction of myoclonic, clonic, and tonic seizures were expressed as means ± SD and compared with one-way ANOVA.

**Microglia: gene expression and viability analyses**. Primary microglia culture and treatment: Primary microglia were collected from P7 brain of OF1 strain mice (Charles River, L'Arbresle, France). Microglia was obtained after cell dissociation and CD11B+ cell separation using Neural Tissue Dissociation Kit (P) and anti-CD11B microbeads (MACS Technology), according to the manufacturer's protocol (Miltenyi Biotec, Germany). Microglia were pelleted in macrophage-serum-free medium (Gibco) supplemented with 1% P/S (Gibco). For RNA-seq experiment, 1 ml/well of cell suspension at $6 \times 10^5$ cells/ml was plated in 12-well culture plates. For immunofluorescence, 250 μL of cell suspension at $3 \times 10^5$ cells/ml was plated in μ-Slide 8 Well Glass Bottom (Ibidi, Biovalley, France). The medium was replaced 24 h later, and experiments started 24 h after replacement of medium. Cells were treated with PLX 3397 1 μM (Selleckchem) or DMSO (final concentration 0.05%) for 30 min before adding PBS or mouse interleukin-1β (50 ng/ml) + interferon-γ (20 ng/ml) (R&D Systems). Twenty-four hours later, media were removed and plates frozen at −80 °C for RNA-seq experiment. For immunocytofluorescence one positive control was added to other conditions using 4 h of incubation with staurosporine 1 μM (Abcam). Cells were fixed at room temperature with 4% formaldehyde for 20 min.

RNA extraction, quantification, and quality: Total RNA from primary microglia was extracted using NucleoSpin RNA XS according to the manufacturer's instructions (Macherey-Nagel, France). RNA concentration and quality were assessed by spectrophotometry with the NanodropTM apparatus (ThermoFisher Scientific, MA, USA) and Experion RNA StdSens Analysis Kit (Bio-Rad).

RNA-seq: mRNA was sequenced on Illumina HiSeq 4000 at an average read depth of 40 million reads per sample.

Immunocytofluorescence and quantification: Immunofluorescence staining on ibidi chambers was performed at restroom temperature and started by 30 min of

incubation in a blocking buffer (PBS/3% bovine serum albumin (BSA)/0.3% Triton X-100) followed by 2 h of incubation with biotinylated lycopersicon esculentum (Tomato) lectin (1:500, Vector Laboratories) and rabbit anti-cleaved caspase-3 (Asp175) antibody (1:400, Cell Signaling) diluted in PBS/1% BSA/0.3%Triton X-100. After rinsing with PBS, cells were incubated 1 h with streptavidin Cy3 (1:1000, ThermoFisher Scientific) and donkey anti-rabbit Alexa 488 (1:1000, ThermoFisher Scientific) followed by a counterstaining with DAPI.

Fives pictures per well were made using Nikon Eclipse Ti-E microscope (×10) and fields were automatically defined by the software not by the experimenter. An experimenter blind to treatment group performed all analyses. All cells were Tomato lectin positive (TL+). The number DAPI+ nucleus was counted as well as the cleaved caspase-3+ cells using the Fiji Software. The number of DAPI+ nucleus and cleaved caspase-3+ cells were expressed per mm$^2$.

**Microglia: phenotype and viability analyses**. Primary microglia cell culture: Forebrains were isolated from post-natal day 7–8 (P7–8) mice (C57BL/6J) and meninges were carefully removed. Brains were dissociated using the Papain Dissociation System (Worthington) according to the manufacturer's instructions. Homogenates were filtered through a 40 μm cell strainer (Falcon) and re-suspended in complete medium: DMEM GlutaMAX (ThermoFisher Scientific) supplemented with 10% fetal bovine serum and 1% penicillin/streptomycin (P/S) (ThermoFisher Scientific). Single-cell suspensions were then transferred into T75 flasks and incubated at 37 °C in 5% $CO_2$ for 1 week. Microglia were isolated from mixed glial cell cultures by shaking flasks for 1 h at 200 rpm at 37 °C, re-suspended in complete medium with 20 ng/ml M-CSF (R&D Systems) and grown for 7 days in 2-well culture insert 24-well (Ibidi) or 96-well (Greiner) plates.

Scratch wound migration assay: Primary microglia were seeded at a density of 30,000 cells per insert in 2-well culture insert 24-well plates (Ibidi). Cells were incubated at 37 °C in 5% $CO_2$ until reaching approximately 80% confluence. Culture inserts were then carefully removed followed by washing of the cell monolayer with fresh complete medium and imaging of the scratch area using an EVOS digital inverted light microscope. Primary microglia were treated with 1 μM of PLX3397 for 24 h and the scratch area re-imaged. Microglia cell migration into the scratch area was quantified using ImageJ.

LDH assay: Cell death activity in brain slice culture was measured in slice-bathing fluids collected 4 h after exposure to zymosan beads. The LDH Activity Colorimetric Assay Kit (BioVision) was used following the manufacturer's instructions.

Live/dead assay: Primary mouse microglia isolated and cultured as described above were exposed to vehicle (medium + 0.01% DMSO) or 1 μM PLX3397 for 24 h at 37 °C at 4% $CO_2$. Cell viability was measured using a LIVE/DEAD Viability/Cytotoxicity Kit as per the manufacturer's instructions (ThermoFisher Scientific). Labeled cells were fixed and imaged using a Zeiss AxioObserver Z1 fluorescent microscope (Zeiss).

Ex vivo phagocytosis assay using acute brain slices: After exposure to isoflurane, PLX3397-treated pilocarpine mice were decapitated and brains rapidly dissected. Sagittal sections were generated at 300 μm thickness using a Leica VT 1200S vibratome (Leica) in ice-cold carbogen (95% $CO_2$, 5% $O_2$)-bubbled artificial cerebrospinal fluid solution (aCSF) consisting of 126 mM choline chloride, 3 mM KCl, 2.4 mM $CaCl_2$, 1.3 mM $MgCl_2$, 26 mM $NaHCO_3$, 1.24 mM $NaH_2PO_4$, and 10 mM glucose. Slices were immediately placed for 1 h at 35 °C in incubation chambers (Prechamber BSC-PC, Harvard Apparatus) filled with carbogen-bubbled aCSF (with choline chloride replaced by a stoichiometric amount of NaCl). Slices were then incubated for 1 h at 37 °C followed by incubation with pH Rodo-conjugated zymosan bioparticles and prepared as per the manufacturer's instructions (ThermoFisher Scientific) at 37 °C for 1 h. After washing slices with PBS, they were fixed in 4% paraformaldehyde for 1 h at room temperature before immunostaining. Fixed slices were incubated for 3 h with blocking solution (normal goat serum 5%, 0.05% Triton X in PBS) and incubated for 48 h with primary antibody: anti-Iba1 (Synaptic Systems, 234004). Slices were washed 3× for 15 min in PBS and incubated with secondary antibody (anti-guinea pig IgG, Alexa 488; ThermoFisher Scientific) for 3 h at room temperature. After washing in 1× PBS, slices were counterstained with DAPI and mounted. Images were acquired using the Zeiss LSM880 confocal microscope (Zeiss) and the number of particles engulfed by Iba1 + microglia and the number of phagocytic microglia counted.

Microglia morphology analysis: Multiple Z-stack images (×63 objective, Zeiss LSM880) were captured from random areas of vehicle-treated and PLX3397-treated pilocarpine brain sections and processed using the Zeiss ZEN 2.3 software (Zeiss). These images were then used for microglia morphology analysis using the ImageJ plugin *NeurphologyJ Interactive* to measure microglial soma number/area and process number/area.

Statistical analysis: Results are presented as means ± standard error of the mean (SEM). Statistics were calculated using the GraphPad Prism 7 using two-tailed Student's *t* test.

**PLX3397 PKs in mouse plasma and brain**. PLX3397 at 30 mg/kg (p.o.) was administered in naïve male NMRI (Naval Medical Research Institute) mice ($n = 2$–4). Plasma samples were taken at 0.25, 0.5, 1, 2, and 6 h after dosing, while terminal plasma and brain samples were collected 24 h after dosing. In addition, pilocarpine NMRI mice ($n = 8$) were dosed at 3 and 30 mg/kg (p.o.), once a day for

1 week, and terminal plasma and brain samples were taken 24 h after the last administration and analyzed by liquid chromatography with tandem mass spectrometry bioanalytical method. The relationships between peak area ratio (analyte/IS) and plasma concentration was linear over the range of 20–50,000 ng/ml for PLX3397. The relationships between peak area ratio (analyte/IS) and brain concentration was linear over the range of 100–250,000 ng/g for PLX3397. Data are submitted to a $1/x^2$ weighing factor and UCB30542 is used as an internal standard for PLX3397. Raw data were smoothed (no. of smooths: 3), peaks were automatically integrated and integrations were checked individually.

## Data availability

The RNA-seq data generated in this study (a) to build gene co-expression modules and (b) to assess effect of PLX3397 on the microglial transcriptome have been submitted to the European Nucleotide Archive (ENA) under accession numbers PRJEB18790 and PRJEB27765 (https://www.ebi.ac.uk/ena/data/view/PRJEB18790 and https://www.ebi.ac.uk/ena/data/view/PRJEB27765), respectively. The gene expression data generated to assess the transcriptional effects of PLX3397 in the pilocarpine model of epilepsy has been submitted to NCBI Gene Expression Omnibus (GEO) under accession number GSE77578 (https://www.ncbi.nlm.nih.gov/geo/query/acc.cgi?acc=GSE77578).

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

## Acknowledgements

We acknowledge funding from UCB Pharma, Imperial College NIHR Biomedical Research Center (BRC) Scheme, the Singapore Ministry of Health, and the European Union's Seventh Framework Programme (FP7/2007–2013) under grant agreement number 602102 (EPITARGET, to E.P. and M.R.J.). We thank Nathalie Leclère (ELISA and LDH assays) and Gregory Szczesny (development of Ictal Analysis Software).

## Author contributions

M.R.J., R.M.K., and E.P. conceived and designed the study and acquired funds. M.R.J., R.M.K., E.P. P.K.S., J.v.E., A.D.-D., J.V.S., J.B., and P.G. performed data analysis, and developed and implemented the methodology. Data generation, curation, and/or laboratory experiments were performed by M.M., P.K.S., J.v.E., J.V.S., P.G., I.K., J.K., J.G., G.G., L.L., A.D.-D., M.M., B.D., C.V., P.F., K.L., G.M.-C., I.N., S.-A.C., A.C., K.S., and F.V. R.M.K., E.P., M.R.J., P.K.S., and J.v.E. wrote the manuscript and all authors contributed, read, and approved the final version.

## Additional information

**Competing interests:** J.v.E., P.G., M.M., J.V.S., B.D., C.V., P.F., K.L., G.M.-C., A.C., F.V., I.N., J.K., J.G., G.G., S.-A.C., I.K., and R.M.K. are employees of UCB Pharma. M.R.J. and E.P. have received research funding from UCB Pharma. The authors declare no other competing interests.

