## [Peer Review File · Nature Communications]

Editorial Note: this manuscript has been previously reviewed at another journal that is not operating a transparent peer review scheme. This document only contains reviewer comments and rebuttal letters for versions considered at *Nature Communications*.

REVIEWERS' COMMENTS:

Reviewer #2 (Remarks to the Author):

Srivastava and colleagues describe a new framework to predict membrane receptor drug targets and validate the proof-of-concept in pre-clinical models of epilepsy. The authors begin with a major RNAseq analysis of differentially expressed genes in the hippocampus of epileptic mice. They then cluster these based on co-expression relationships into various "modules". GO analysis identified immune-related functions as highly enriched which they subsequently used data on cell-specific gene expression to link to microglia. A set of the modules showed associations with seizure frequency in the epileptic mice, of which one – module 18 – was the most correlated with seizure frequency. They next looked for surface-expressed receptors likely to regulate the gene networks in the module, linking membrane receptors to transcription factors and their target genes. From these, the best hit was M-CSF (Csf1R). This has not previously been linked to epilepsy. They then took advantage of a small molecule inhibitor of Csf1R (PLX3397) to show that dosing mice with the compound restores expression of genes in the module 18 network. PLX3397 had anti-seizure effects in two in vivo mouse models (pilocarpine and intrahippocampal kainate) and an in vitro organotypic slice culture model. Importantly, and in line with predictions, the drug did not have acute anti-seizure effects in models where seizures are evoked in normal mice. Overall the authors are to be commended on an innovative approach, data-rich and potentially important study that lays out a logical bioinformatics approach that led to a new druggable target for seizure control. A similar approach could certainly lead to new targets for other diseases.